# Prednisolone prescribing practices for dogs in Australia

**Bonnie Purcell**[1]*, **Anke Wiethoelter**[1], **Julien Dandrieux**[1,2]

**1** Melbourne Veterinary School, Faculty of Science, The University of Melbourne, Parkville, Victoria, Australia, **2** Hospital for Small Animals, Royal (Dick) School of Veterinary Studies, College of Medicine and Veterinary Medicine, University of Edinburgh, Easter Bush, Midlothian, United Kingdom

* purcellbon@gmail.com

**Data Availability Statement:** All relevant data are within the manuscript and its Supporting Information files.

**Funding:** The author(s) received no specific funding for this work.

## Abstract

Although prednisolone is a routinely prescribed medication in dogs, there is a lack of information regarding prednisolone prescribing practices by veterinarians. This study aims to describe characteristics of dogs receiving prednisolone, disease processes treated, doses prescribed as well as to identify factors influencing the dose rate in Australia. The VetCompass Australia database was queried to identify dogs prescribed prednisolone between 1 July 2016 to 31 July 2018 (inclusive). A random sample of 2,000 dogs from this population were selected. Dog demographic data, prednisolone dose and indication for prescription were collated. Indicated dose for the condition treated was compared to prescribed dose. Multivariable linear regression was used to identify patient-level characteristics associated with prescribed prednisolone dose. A large and small breed dog cohort, treated for the same disease process, were compared for differences in dosing. Median age of dogs was 73 (range 2 to 247) months and median body weight was 17 (range 1.56 to 90) kg. Median prescribed prednisolone dose was 0.8 mg/kg/day, with most dogs receiving an anti-inflammatory dose (0.3–1 mg/kg/day, 58%). Prednisolone prescriptions were predominantly for diseases of the integument ($n = 1645$, 82%) followed by unknown indication and respiratory disease. A total of 152 dogs (8%) were prescribed immunosuppressive doses of prednisolone for conditions where an anti-inflammatory dose would be recommended. Increases in bodyweight were associated with lower doses on mg/kg scale but higher doses on a mg/m$^2$ scale ($p < 0.001$). Overall, prednisolone was primarily used as an anti-inflammatory in this population, with some inappropriate use of immunosuppressive doses. Increasing bodyweight was associated with a small reduction in dose in mg/kg, suggesting that clinicians are adjusting prednisolone dose rates based on dog bodyweight.

## Introduction

Prednisolone is a medication commonly used by companion animal veterinarians [1]. Prednisolone can be used for physiologic corticosteroid replacement, as an anti-inflammatory and as a first-line immunosuppressant [2]. Dosage recommendations are guided by intent for use,

**Competing interests:** The authors have declared that no competing interests exist.

with the lowest dose rates used for physiological replacement and the highest dose rates for immunosuppression [2, 3].

In dogs, as in other domestic species, the dose of prednisolone administered is determined by bodyweight [2]. For physiologic replacement to treat hypoadrenocorticism the recommended dose rate is 0.1 to 0.3 mg/kg/day [2, 4]. Anti-inflammatory dose rates are generally accepted to be in the range of 0.5 to 1.0 mg/kg/day [2]. Immunosuppressive doses are usually reported to be 2 to 4 mg/kg/day and implemented as a course which is tapered-off over a number of weeks [2, 5]. More recently, immunosuppressant doses have been recommended to be calculated using body surface area (body surface area in meters$^2$ = 10.1 X (weight in grams)$^{2/3}$) ÷ 10,000) for dogs greater than 25 kg, with doses not exceeding 50 to 60 mg/m$^2$/day due to the perceived increased risk of adverse effects in larger dogs [2, 5]. Dosage interval varies from 12 hourly to every other day or less frequently during the tapering-off period [2, 5]. While dose guidelines for using prednisolone as an anti-inflammatory as opposed to an immunosuppressant exist, the differentiation between is somewhat arbitrary [6]. It is likely that individual veterinarians exercise discretion in the amount of prednisolone administered for a given dog with a given disease condition.

Several studies from the United Kingdom have reviewed glucocorticoid, including prednisolone, use in small animal practice [1, 7, 8]. Information gathered included descriptions of the frequency of glucocorticoid prescriptions, signalment characteristics of the dogs that were prescribed glucocorticoids, risk factors for being prescribed glucocorticoids and median dose rates [1, 7, 8]. These studies provide some information about prednisolone prescribing practices but are specific to the United Kingdom. Furthermore, these studies provide no information on the frequency of use of prednisolone for physiologic, anti-inflammatory, and immunosuppressive indications. To the best of our knowledge, there are no studies investigating prednisolone dosing regimens for dogs presented to primary care practices in Australia.

With this background, the aims of this study were to: (1) To describe the characteristics of dogs receiving prednisolone in Australia; (2) To quantify prednisolone dose rates (in mg/kg and mg/m$^2$) and the frequency of prednisolone administration for physiologic, anti-inflammatory and immunosuppressive uses; (3) To describe the disease processes treated with prednisolone in this population; (4) To quantify the frequency of inappropriate use of high doses of prednisolone; (5) To determine if any factors influence prednisolone dose rate and if larger dogs are more likely to receive a lower dose of prednisolone, compared to smaller dogs, for the same indication.

## Materials and methods

This was a cross-sectional study using individual animal clinical records through VetCompass Australia (Human Ethics Project Title: VetCompass Australia. Project number: 2013/919. The University of Sydney). VetCompass Australia is a national small animal surveillance system that collects de-identified clinical records from contributing primary care practices across Australia [9]. At the time of data retrieval, 137 (general and referral) practices contributed medical records to the VetCompass database.

The VetCompass Australia database was queried to identify dogs prescribed prednisolone containing products for consultation events that occurred between 1 July 2016 and 31 July 2018 (inclusive) using the following search terms: "%pred%", "%cort_sone%", "%steroid%", "%steriod%", "%crolone%", "%solone%", "%panafcortelone%", "%niralone%". The percent symbol was used as a wildcard character representing any number of letters, numbers, spaces and punctuation. This allows terms containing these phrases to be identified. These terms were developed to retrieve prednisolone, as well as product names for oral or injectable

prednisolone containing products available in Australia, and to account for spelling mistakes. Consultation records returning a positive result for the listed search terms ($n$ = 514,759) were checked for eligibility for this study using the inclusion criteria described below.

To be eligible for this study a dog's consultation event had to have a prednisolone product prescribed (oral or injectable), a dose recorded and a recorded body weight at the time of the prescription. Records where a prednisolone product was prescribed were identified from the retrieved dataset by selecting for prednisolone containing products using the pharmaceutical item name field. This yielded 19,412 records where a prednisolone product was prescribed to a dog with a recorded body weight. Consultation records where the bodyweight of the dog was less than 1 kg or greater than 80 kg were reviewed to ensure that the patient signalment matched the recorded bodyweight. Implausible consultation records ($n$ = 5) were excluded. Consultation records where bodyweight was recorded as an estimate ($n$ = 10) were excluded. The dataset was screened for duplicate consultations, and these were excluded after review ($n$ = 19). From these eligible records ($n$ = 19,378), 2,000 medical records were randomly selected using a random number generator (Excel, Microsoft, Redmond, USA) for manual review and data acquisition. If multiple consultation records were returned for the same dog only one of the listed consultation records were selected, at random, for inclusion in the study. The sample size chosen was calculated based on a linear regression model detecting an association between dog variables and the prednisolone dose prescribed, with 80% confidence [10]. For a linear regression model with three explanatory variables, we set the alpha level to declare statistical significance to 0.05, power at 0.80 and (conservatively) assumed that the model would explain 5% of the variance in prednisolone dose. With these assumptions, a minimum of 212 dogs were required to meet the specifications of the study. Given lack of independence in the data arising from dogs clustered within veterinary clinics, and some of the variation in the data arising from veterinary clinic-level effects we (again, conservatively) assumed a design effect in the order of 10, increasing our required sample size to 2,000 [11].

For each included consultation record ($n$ = 2,000) the patient's age, sex and neuter status, breed bodyweight and prednisolone starting dose (in mg) were retrieved. Patient breeds were categorised using the Australian National Kennel Council classes (toy, terrier, gundog, hound, working dog, utility and non-sporting) [12]. If the breed listed in the patient record was either not recorded or not consistent with the Australian National Kennel Council breed classes, the dog's breed was recorded as 'other'. Bodyweights were categorised into weight classes; less than 25 kg, 25 kg to less than 40 kg and greater than or equal to 40 kg.

Starting dose was defined as the dose the patient was prescribed to receive initially before any tapering occurred. If dosing was every second or every third day, this was averaged into a daily dose. Body surface area in m$^2$ was calculated using the formula $0.101 \times$ bodyweight (kg)$^{2/3}$. Prednisolone dose rate was calculated in both mg/kg and mg/m$^2$ for each consultation record. Dose categories were classified as physiologic ($< 0.3$ mg/kg/day), anti-inflammatory (0.3 to 1.0 mg/kg/day), intermediate ($> 1.0$ to $< 1.5$ mg/kg/day) or immunosuppressive ($\geq 1.5$ mg/kg/day). The intermediate dose category was used to allow for a gap between anti-inflammatory and immunosuppressive doses and to account for adjustments due to tablet sizes. For dogs equal to or greater than 25 kg we created an additional category for doses $\geq 50$ mg/m$^2$/day (immunosuppressant dose for body surface area dosing).

Indication for prescription was reviewed by the first author (BP) and categorised according to body system (see S1 Appendix for body system categories). To define the body system involved the attending veterinarian's primary suspected diagnosis was used. If no diagnosis was listed, body system was assigned based on the presenting clinical complaint. If more than one reason for prescription was provided, the primary reason was used, based on either review of the attending veterinarian's notes made at the time of each consultation or the presenting

clinical complaint. If there was no clear rationale for use following review of the consultation records, the indication was classified as 'unknown'.

The dose prescribed was also compared to the dose indicated for the condition being treated (see S2 Appendix for indicated dose categories for specific indications or diseases). Specifically, dosing in the immunosuppressive dose range (equal to or greater than 1.5 mg/kg/day for all dogs, or greater than or equal to 50 mg/m$^2$/day for dogs over 25 kg) for an inflammatory condition, was considered an inappropriately high dose. Inappropriately low doses were not evaluated for, as animals on a tapering dosage could not be retrospectively accounted for. No attempt was made to assess the validity of the listed diagnosis based on the consultation record. The indication for prescription was described in more detail for dogs receiving higher immunosuppressive doses of $\geq$2.0 mg/kg/day.

To assess the effect of weight on dosing protocol, we compared the prednisolone dosage used to treat Maltese versus Labradors for inflammatory skin disease. These two breeds were selected as frequently present in the database and representative of a small and large dogs group treated for the same disease category.

## Statistical analyses

Dog demographic and consultation record details were described using descriptive statistics. Continuous variables were tested for normality using the Shapiro Wilk test. Continuous variables were described using means and standard deviations as well as median and range, for clarity. Categorical variables were summarised using frequency tables.

A linear regression model was developed to quantify the association between dog and consultation record variables and prednisolone dose expressed on either a mg/kg or mg/m$^2$ basis. Univariable linear regression analyses were carried out to identify candidate explanatory variables for multivariable modelling. Dog and consultation record variables associated with prednisolone dose with a p value of <0.25 in the univariable linear regression analyses were carried forward for multivariable modelling analyses. Candidate explanatory variables were first tested for collinearity using the Spearman rank correlation coefficient for categorical variables and the Pearson correlation coefficient for continuous variables. If the calculated correlation coefficient for two variables was greater than 0.6 or less than -0.6, the most clinically meaningful variable of the two was selected to be carried forward for multivariable regression modelling. Explanatory variables for the multivariable model were selected using a backward stepwise approach where all candidate explanatory variables were entered into the multivariable model. Explanatory variables were then removed from the model one at a time, beginning with the least significant, until all variables that remained in the model were significant at p <0.05. The results of the multivariable model for the continuously distributed explanatory variables were reported in terms of the point estimate (and their 95% confidence intervals) of the effect of a stated number of units change in the variable on daily prednisolone dose. For explanatory variables expressed on a categorical scale the results were expressed in terms of the effect of the level of a given variable on daily prednisolone dose, compared with a reference category. Frequency histograms of the residuals from the multivariable model and plots of the residuals versus predicted values were constructed to check that the assumptions of normality and homogeneity of variance had been met. Cook's distance statistics were calculated to identify individual dog prednisolone dosage records that influenced the estimated regression coefficients from the multivariable model.

For the Maltese and Labradors with inflammatory skin disease, the Mann-Whitney U test was used to compare continuous variables, using ranks for age and weight and medians for prednisolone dose data. Pearson's chi squared test was used to compare sex and neuter status.

**Table 1. Demographics of dogs (n = 2,000) prescribed prednisolone in Australian veterinary practices and dose prescribed.**

| Variable | n | Median (Q1, Q3) | Min, max | Mean ± standard deviation | Missing |
|---|---|---|---|---|---|
| Age (months) | 1998 | 73 (34, 116) | 2, 247 | 77 ± 50 | 2 |
| Bodyweight (kg) | 2000 | 17 (8.4, 27.5) | 1.6, 90 | 19 ± 13 | |
| Dose (mg) | 2000 | 10 (5, 20) | 0.36, 140 | 17 ± 14 | |
| Dose (mg/kg/day) | 2000 | 0.8 (0.6, 1.1) | 0.03, 5 | 0.9 ± 0.5 | |
| Dose (mg/m²/day) | 2000 | 21 (13.4, 28.4) | 0.92, 107 | 23 ± 13 | |

Statistical analyses were carried out using Microsoft Excel (Microsoft, Redmond, USA) and IBM SPSS Statistics (IBM Corp, Version 27).

## Results

Descriptive data for the study population (*n* = 2,000) is presented in Tables 1 and 2, with continuous variables in the former and categorical in the latter. The median age at the time of consultation was 73 (range 2 to 247) months. Median bodyweight was 17 (range 1.6 to 90) kg with most dogs under 25 kg (68%). Most dogs were neutered, with a slight predominance of males. The most common breeds were Staffordshire bull terriers (*n* = 278), followed by Maltese (*n* = 166), Labradors (*n* = 128), Jack Russell terriers (*n* = 93), all types of poodles (*n* = 75) and border collies (*n* = 75).

Data describing prednisolone prescriptions are detailed in Tables 1 and 3. The median dose prescribed was in the anti-inflammatory range (0.8 mg/kg/day, anti-inflammatory dose 0.5–1 mg/kg/day [2]) with doses ranging from low physiologic doses up to 5 mg/kg/day (Fig 1). Dose prescribed were categorised by dose range, with most dogs being prescribed an anti-

**Table 2. Sex-neuter status, breed and body weight category of dogs (n = 2,000) prescribed prednisolone in Australian veterinary practices.**

| Variable | n (%) |
|---|---|
| Sex-neuter status: | |
| Female entire | 121 (6) |
| Female neutered | 793 (40) |
| Male entire | 206 (10) |
| Male neutered | 880 (44) |
| Breed category: | |
| Terrier | 490 (24) |
| Toy | 355 (18) |
| Working | 262 (13) |
| Non-sporting | 258 (13) |
| Gun dog | 234 (12) |
| Utility | 170 (8) |
| Hound | 96 (5) |
| Other* | 135 (7) |
| Weight category: | |
| <25 kg | 1360 (68) |
| 25 to 40 kg | 515 (26) |
| ≥40 kg | 125 (6) |

* 'Other': the breed listed in the patient record was either not recorded or not consistent with the Australian National Kennel Council breed classes

**Table 3. Characteristics of prednisolone prescriptions in 2,000 dogs visiting Australian veterinary practices.**

| Variable | mg/kg/day | n (%) |
|---|---|---|
| Dose category: | | |
| Physiologic | < 0.3 | 136 (7) |
| Anti-inflammatory | 0.3–1.0 | 1169 (58) |
| Intermediate | > 1.0 - < 1.5 | 491 (25) |
| Immunosuppressive | ≥ 1.5 | 204 (10) |
| Indication: | | |
| Integument | | 1645 (82%) |
| Unknown | | 104 (5%) |
| Respiratory | | 82 (4%) |
| Neurological | | 63 (3%) |
| Haematopoietic | | 36 (2%) |
| Ocular | | 24 (1%) |
| Gastrointestinal | | 23 (1%) |
| Endocrine | | 10 (<1%) |
| Musculoskeletal | | 7 (<1%) |
| Cardiovascular | | 6 (<1%) |
| Dosing formula: | | |
| Intravenous | | 1 (0.05%) |
| Oral (tablet) | | 1993 (99.65%) |
| Oral (liquid) | | 6 (0.3%) |

inflammatory dose ($n = 1169$, 58%). Physiologic and immunosuppressive doses were prescribed to 7% ($n = 136$) and 10% ($n = 204$) of dogs, respectively.

Most prescriptions ($n = 1,645$, 82%) were for diseases affecting the integument, with all other indications being individually less than 10% of prescriptions (Table 3). Indication for prescription could not be determined for 5% ($n = 104$) of dogs. Immune mediated diseases included suspected immune mediated haemolytic anaemia (IMHA, $n = 10$), immune mediated thrombocytopenia (IMTP, $n = 5$), concurrent IMHA and IMTP ($n = 1$) and immune mediated polyarthritis ($n = 3$). Fifteen dogs were treated for lymphoma. Dogs prescribed prednisolone within the "endocrine" body system category included eight dogs with hypoadrenocorticism and two dogs treated for hypoglycaemia. The dogs with hypoadrenocorticism had a median dose of 0.21 (range 0.13 to 0.75) mg/kg/day.

The product names of prednisolone containing oral or injectable products used in the study population of dogs are listed in S3 Appendix and the number of dogs receiving different types of formulations (tablets, oral liquid or injectable) in Table 3. Almost all prescriptions were for oral tablet formulas ($n = 1,993$, 99.7%). One dog received intravenous prednisolone.

Comparisons of the prescribed dose with the indicated dose are summarised in Table 4. An anti-inflammatory dose was prescribed for an inflammatory condition in 55% of dogs ($n = 1,096$). Use of an immunosuppressive dose (equal to or greater than 1.5 mg/kg/day), for an inflammatory condition occurred in 8% of dogs ($n = 152$) and was considered inappropriate. There were 640 dogs who were 25 kg in bodyweight or greater, with 4% ($n = 27$) prescribed immunosuppressive doses based on body surface area dosing (50 mg/m$^2$ per day or greater). For dogs over 25kg receiving doses of 50 mg/m$^2$/day or greater, 67% ($n = 18$) were prescribed for conditions where an anti-inflammatory dose was indicated. These 18 prescriptions only accounted for 3% of all prescriptions for dogs 25 kg or greater. Fifteen of these were prescribed immunosuppressive dosages for inflammatory skin or ear disease, one dog for urticaria, one

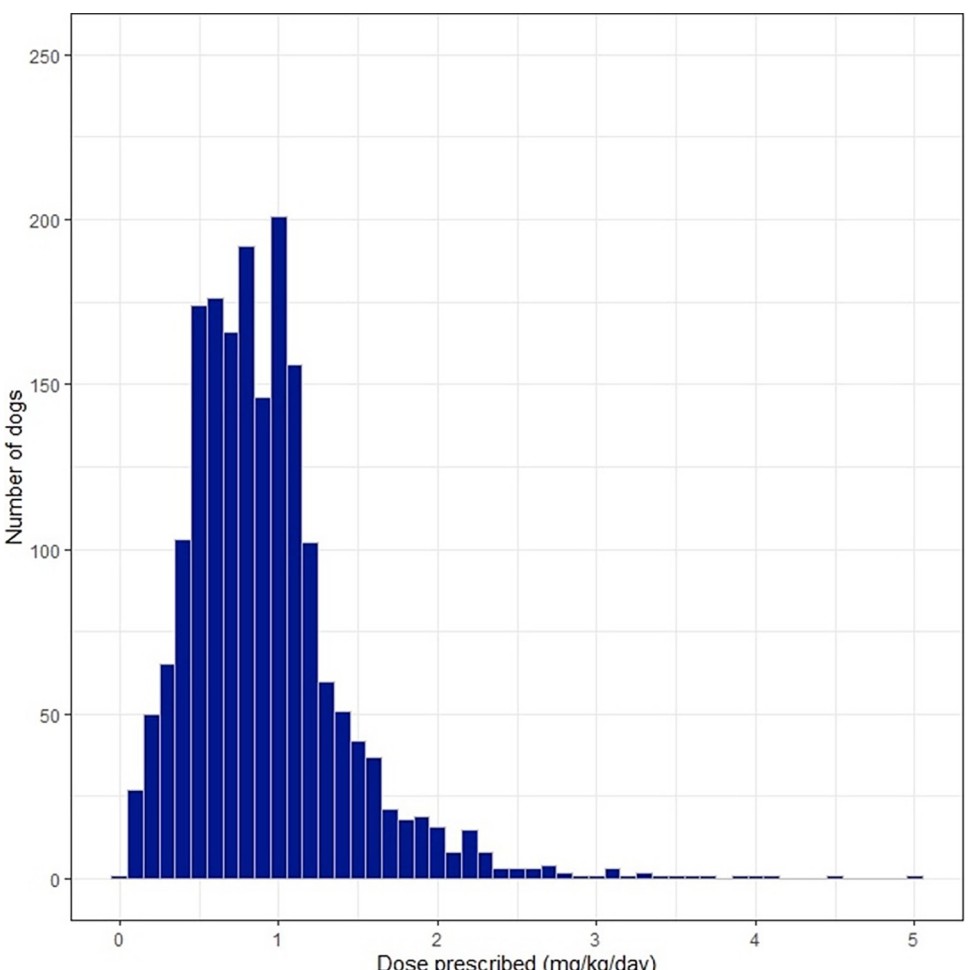

**Fig 1. Starting dose of prednisolone prescribed in mg/kg/day.**

dog for an inflamed mammary mass and one dog for degenerative joint disease. Eight dogs were prescribed appropriately for conditions where an immunosuppressive dose was warranted, for one dog the reason for prescription was unclear. Dogs receiving 2 mg/kg/day or greater of prednisolone ($n = 77$) were treated for the conditions listed in Table 5. Of the 77 consultations, 28 (36%) were for apparently uncomplicated inflammatory skin or ear conditions.

**Table 4. Comparison of prescribed prednisolone dose (mg/kg) to indicated dose to 2,000 Australian dogs.**

| Dose prescribed | Indicated dose | | | |
|---|---|---|---|---|
| | Physiologic | Anti-inflammatory | Immunosuppressive | Unknown |
| All dogs: | | | | |
| < 0.3 mg/kg | 7 | 99 | 8 | 22 |
| 0.3–1.0 mg/kg | 3 | 1096 | 15 | 55 |
| > 1.0 - < 1.5 mg/kg | | 454 | 19 | 18 |
| ≥ 1.5 mg/kg | | 152 | 38 | 14 |
| Dogs ≥ 25 kg: | | | | |
| < 50 mg/m²/day | 1 | 564 | 19 | 29 |
| ≥ 50 mg/m²/day | 0 | 18 | 8 | 1 |

**Table 5. Indication for prednisolone in Australian dogs receiving ≥2mg/kg/day of prednisolone (*n* = 77).**

| Body system | Primary differential or clinical indication | Number of dogs |
|---|---|---|
| Integument | Inflammatory skin or ear disease | 28 |
| | Lupus erythematosus | 2 |
| | Urticaria | 1 |
| | Mammary carcinoma with dermatitis | 1 |
| | Uncharacterised dermopathy (vesicles) | 1 |
| | Sterile nodular panniculitis | 1 |
| | Ear mass | 1 |
| | Cutaneous mass | 1 |
| Neurological | Meningitis | 5 |
| | Steroid responsive cerebellitis | 2 |
| | Neck or spinal pain, suspected disk disease | 2 |
| | Seizures | 1 |
| | Steroid responsive meningitis arteritis | 1 |
| | Paresis and spinal pain (no differential) | 1 |
| | Acute blindness | 1 |
| Haematopoietic | Immune mediated haemolytic anaemia | 6 |
| | Lymphoma | 3 |
| | Immune mediated thrombocytopenia | 2 |
| | Concurrent immune mediated haemolytic anaemia and immune mediated thrombocytopenia | 1 |
| | Chronic leukaemia | 1 |
| Respiratory | Upper airway obstruction | 3 |
| | Sneezing | 2 |
| | Coughing | 1 |
| | Epistaxis and suspected nasal neoplasia | 1 |
| Unknown | | 4 |
| Gastrointestinal | Vomiting, suspected neoplasia | 1 |
| | Oral mass lesion | 1 |
| Ocular | Trauma and secondary hyphaemia | 1 |
| Musculoskeletal | Immune mediated polyarthritis | 1 |

Univariable linear regression analysis identified age, sex/neuter status, bodyweight, BSA, weight class, body system and indicated dose to have a statistically significant ($p < 0.25$) association with daily prednisolone dose in mg/kg. These factors, along with breed class, were also identified to be significant for dose expressed in $mg/m^2$. These variables were carried forward to the multivariable linear regression analysis. As weight, BSA and weight class were strongly co-linear, weight (kg) was used in the multivariable analyses.

Tables 6 and 7 present the results of the multivariable linear regression models for prednisolone dose in mg/kg and $mg/m^2$ (respectively). Bodyweight, age, sex/neuter status, body system and indicated dose remained statistically significant ($p < 0.05$) in the mg/kg model. Bodyweight, age, body system and indicated dose were statistically significant in the $mg/m^2$ model. Endocrine (as a body system) was a redundant category as this group of dogs was identical to the dogs in the physiologic dose group in the indicated dose category. Increasing bodyweight was associated with lower doses in mg/kg, but higher doses in $mg/m^2$. Increasing age was associated with a slight reduction in dose in both mg/kg and $mg/m^2$. Conditions requiring anti-inflammatory or physiologic doses were associated with lower doses compared to immunosuppressive conditions. Some body systems were also associated with higher doses in mg/kg and $mg/m^2$, namely haematopoietic and neurologic conditions, compared to integument as a

**Table 6. Multivariable analysis for variables associated with prednisolone dose in mg/kg.**

| Variable | Number of dogs | Unstandardised regression coefficient (Standard Error) | P (likelihood ratio) | 95% Wald confidence interval |
|---|---|---|---|---|
| Intercept | | 1.468 (0.0804) | <0.001 | 1.311–1.626 |
| Age (years) | 1886 | -0.009 (0.0029) | 0.003 | -0.014 – -0.003 |
| Weight (kg) | 1886 | -0.005 (0.0009)[a] | <0.001 | -0.007 – -0.003 |
| Sex | | | 0.024 | |
| Male neutered | 883 | Ref | | |
| Female entire | 114 | 0.145 (0.0487) | | 0.050–0.241 |
| Female neutered | 736 | 0.011 (0.0246) | | -0.037–0.059 |
| Male entire | 203 | 0.042 (0.0384) | | -0.034–0.117 |
| Body system | | | <0.001 | |
| Integument | 1643 | Ref | | |
| Ocular | 24 | 0.111 (0.0996) | | -0.085–0.306 |
| Respiratory | 80 | 0.093 (0.0573) | | -0.020–0.205 |
| Gastrointestinal | 22 | 0.270 (0.1047) | | 0.064–0.475 |
| Neurological | 61 | 0.293 (0.0670) | | 0.161–0.424 |
| Haematopoietic | 33 | 0.489 (0.1116)[b] | | 0.270–0.707 |
| Endocrine | 10 | -1.019 (0.1708) | | -1.354 – -0.684 |
| Musculoskeletal | 7 | 0.036 (0.1880) | | -0.333–0.404 |
| Cardiovascular | 6 | 0.188 (0.1981) | | -0.201–0.576 |
| Indicated dose | | | <0.001 | |
| Immunosuppressive | 80 | Ref | | |
| Physiologic | 10 | * | | |
| Anti-inflammatory | 1796 | -0.461 (0.0752) | | -0.608 – -0.313 |

Alkaike's information criterion: 2645.326

Ref: Reference category

*Set to zero as value redundant as all dogs in the endocrine group are shared with the physiologic group

[a] Interpretation: An increase in body weight by 1kg is independently associated with a reduction in prednisolone dose by 0.005mg/kg/day (95% CI: -0.007 to -0.003)

[b] Interpretation: Compared to the reference category (integument), treatment for a disease condition in the haematopoietic category is independently associated with an increase in prednisolone dose by 0.489mg/kg/day (95% CI: 0.270–0.707)

reference. Female entire status was associated with higher doses compared to the male neutered category for dose in mg/kg, but not mg/m$^2$. Residuals for both models plotted as frequency histograms were consistent with a normal distribution. A scatterplot of model residuals as a function of prednisolone dose predicted by the model showed no evidence of heteroskedasticity. Removal of individual dog prednisolone dosage records where Cook's distance was greater than 0.05 resulted in no biologically meaningful change in the estimated regression coefficients. Therefore, those records with Cook's distance greater than 0.05 were retained in the final models.

Dog demographic data and prednisolone doses for Maltese and Labradors with inflammatory skin disease are shown in Table 8. Median bodyweights were markedly different: 7.6 kg for the Maltese and 33 kg for the Labradors, allowing them to act as a small breed and large breed group for dosage comparisons. Prescribed dose of prednisolone differed by breed category with the Maltese group prescribed a higher dose in mg/kg whereas the Labrador group were prescribed a higher dose in mg/m$^2$. The groups also differed in terms of age and sex/neuter status, with the Maltese group comprised of a higher proportion of older male dogs compared with the Labrador group.

Table 7. Multivariable analysis for variables associated with prednisolone dose in mg/m$^2$.

| Variable | Number of dogs | Unstandardised regression coefficient (Standard Error) | P (likelihood ratio) | 95% Wald confidence interval |
|---|---|---|---|---|
| Intercept | | 29.489 (1.9592) | <0.001 | 25.649–33.329 |
| Age (years) | 1886 | -0.241 (0.0712) | <0.001 | -0.380 –-0.101 |
| Weight (kg) | 1886 | 0.271 (0.0226) | <0.001 | 0.227–0.316 |
| Body system | | | <0.001 | |
| Integument | 1643 | Ref | | |
| Ocular | 24 | 2.466 (2.4648) | | -2.365–7.297 |
| Respiratory | 80 | 1.657 (1.4181) | | -1.122–4.437 |
| Gastrointestinal | 22 | 4.776 (2.5896) | | -0.299–9.852 |
| Neurological | 61 | 4.556 (1.6540) | | 1.315–7.798 |
| Haematopoietic | 33 | 11.231 (2.7617) | | 5.818–16.644 |
| Endocrine | 10 | -24.155 (4.2257) | | -32.438 –-15.873 |
| Musculoskeletal | 7 | 4.453 (4.6550) | | -4.671–13.576 |
| Cardiovascular | 6 | 3.178 (4.9032) | | -6.432–12.788 |
| Indicated dose | | | <0.001 | |
| Immunosuppressive | 80 | Ref | | |
| Physiologic | 10 | * | | |
| Anti-inflammatory | 1796 | -11.390 (1.8598) | | -15.035 –-7.744 |

Akaike's information criterion: 14744.678

Ref: Reference category

*Set to zero as value redundant as all dogs in the endocrine group are shared with the physiologic group

Table 8. Comparison of Maltese and Labrador dogs treated with prednisolone for inflammatory skin conditions.

| Variable | Maltese (130 dogs) | Labrador (105 dogs) | P value |
|---|---|---|---|
| Sex/neuter[#] | | | 0.001 |
| Female entire | 8 (6%) | 8 (8%) | |
| Female neutered | 37 (28%) | 49 (47%) | |
| Male entire | 6 (5%) | 11 (10%) | |
| Male neutered | 79 (61%) | 37 (35%) | |
| Age (months)* | 87 (3–181) | 60 (6–167) | <0.001 |
| Weight (kg)* | 7.6 (3.4–14.8) | 32.8 (16–53) | <0.001 |
| BSA (m$^2$)^ | 0.38 ± 0.08 | 1.06 ± 0.15 | <0.001 |
| Dose (mg) * | 5 (1–40) | 20 (5–80) | <0.001 |
| Dose (mg/kg) * | 0.99 (0.13–3.88) | 0.73 (0.11–2.22) | 0.003 |
| Dose (mg/m$^2$) * | 17.64 (2.65–83.65) | 22.00 (3.97–72.65) | <0.001 |

BSA: body surface area.

Values reported as either:

[#]Number of dogs (%)

*Median (range)

^Mean +/- standard deviation provided for BSA as normally distributed

P value for sex/neuter refers to outcome of Pearson Chi square test, P value for all other values refers to outcome of Mann Whitney U.

## Discussion

This is the first description of patient demographics, dose prescribed and indication for prescription for dogs receiving prednisolone in veterinary practice in Australia. Prednisolone was prescribed as an anti-inflammatory to most dogs in this population, both in terms of dose prescribed as well as being the most common indication for prescription. Anti-inflammatory doses of prednisolone are in the range of 0.5 to 1 mg/kg/day according to veterinary therapeutic resources [2, 4, 13]. Our anti-inflammatory dose definition was wider (0.3 to 1 mg/kg/day), which may have artificially increased the numbers of dogs in this category. However, we wanted to account for practical limitations of tablet sizing, while still differentiating from lower physiologic replacement doses [4]. The intermediate dose category was created for the purposes of this study (> 1 to < 1.5 mg/kg/day). This was intended to enable differentiation between dogs treated with the widely recognised anti-inflammatory dose versus higher immunosuppressive doses. Some dogs may have received doses in this intermediate category due to limitations of tablet sizing, making it difficult to differentiate use as anti-inflammatory or immunosuppressive intent. There is no robust pharmacodynamic evidence in dogs to justify these different dose categories, though some work has been done to describe the effect of anti-inflammatory doses on canine white blood cell counts [14]. Therefore, we rely on current prescribing guidelines to differentiate these categories of use [2]. With these limitations in mind, this anti-inflammatory dosage range was used for most dogs in this population.

Previous studies have shown use of anti-inflammatory doses of prednisolone to be most common, though with lower medians [7]. O'Neill and colleagues (2012) reported a median dose of 0.53 mg/kg/day in the pilot phase of the United Kingdom VetCompass program [7]. This study was restricted to three veterinary practices, so it may not have been representative of broader prescribing behaviours [7]. Elkolly and colleagues (2020) evaluated dose purely in the context of dogs who had experienced a glucocorticoid side effect. They reported a median starting dose of 0.7mg/kg/day if a glucocorticoid injection was given prior, and 0.52mg/kg/day if no injection was given [8]. Our study evaluated doses in a broader population of dogs, which limits direct comparison, but could suggest use of higher doses of prednisolone in Australia compared to the United Kingdom.

The overwhelming majority of prescriptions in this study population were for disease of the integument. Skin disease has been documented as a risk factor for glucocorticoid prescriptions previously and in a separate study accounted for reason for use in 54% of dogs presented for glucocorticoid related side effects [7, 8]. Skin disease is highly prevalent in Australia, with otitis externa and dermatitis being the two most common diagnoses in a recent study of insured Australian dogs [15]. However, with the development and use of other effective therapies for inflammatory skin disease, such as lokivetmab (Cytopoint, Zoetis Petcare) and oclacitinib (Apoquel, Zoetis Petcare), the frequency of prednisolone use for skin disease may have changed since the study period [16]. Oclacitinib prescription has been associated with a lower odds ratio for prescription of glucocorticoids, as reported by Rynhoud and colleagues (2022) [17].

This study documented inappropriate use of immunosuppressive doses of prednisolone in a small, but not negligible, number of prescriptions (8% of all prescriptions). Of the 77 prescriptions for doses 2 mg/kg/day or greater, 28 of them were for apparently uncomplicated inflammatory skin and ear conditions. Of concern is that 67% of large breed dogs prescribed prednisolone at 50 mg/m²/day or greater had inflammatory disease conditions. The frequency of use of immunosuppressive doses for inflammatory conditions has not been previously evaluated to the best of our knowledge. The impact on individual dogs prescribed these inappropriately high doses is unknown but has the potential to increase the risk of drug induced

morbidity. Corticosteroids, including prednisolone, are not benign drugs and come with the potential for a range of side effects including polyphagia, polydipsia, polyuria, coat quality changes, behavioural changes, sarcopenia, gastrointestinal upset and increased risk of opportunistic infections [18–21]. The occurrence of glucocorticoid induced side effects was 4.9% in a previous study, where the median dose prescribed was anti-inflammatory [8]. Other studies have reported much higher frequencies. A review of clinical trials on oral corticosteroid use in atopic dermatitis found 30–80% of subjects experienced a side effect [22]. Dogs treated for IMHA and IMTP in one study, receiving immunosuppressive doses of glucocorticoids, commonly reported side effects, such as polyuria in 67% of cases [23]. High doses of prednisolone, where not warranted, potentially exposes dogs to more risk of drug induced morbidity. A study evaluating owner reported side effects did demonstrate reductions in polyphagia, polydipsia and polyuria as the dose of prednisolone was reduced over a period of weeks [18].

One of our aims was to evaluate the impact of bodyweight on dosage practices. This was explored in both the linear regression analyses, as well as by comparing prescriptions to Maltese and Labrador dogs for inflammatory skin disease. The latter comparison providing two groups of dogs, treated for the same disease, with markedly different bodyweights. In our multivariable analysis, increases in body weight were associated with an independent, small reduction in prednisolone dose on a mg/kg basis but an increase on a mg/m$^2$ basis. The higher doses in mg/m$^2$ dosing for the larger dogs reflects how increasing weight leads to relatively smaller increases in BSA. Similarly, Labrador dogs received lower doses in mg/kg but higher doses in mg/m$^2$, compared to Maltese. A limitation of this comparison is that the groups did differ in several ways. The Maltese group was significantly older and included more males. Increasing age was associated with reduced prednisolone doses in both mg/kg and mg/m$^2$ in the multivariable analysis, and so would not account for Maltese dogs receiving higher doses in mg/kg. Female entire status specifically was associated with higher doses in the multivariable analysis, and the proportion of female entire dogs between Maltese and Labrador dogs was comparable. There is no clear explanation for female entire dogs receiving higher doses in mg/kg, relative to the male neutered reference category. Given this was not found in the mg/m$^2$ multivariable analysis, it may represent a type 1 error.

The lower dose in mg/kg for larger dogs in this study may indicate that clinicians are being cautious of adverse effects and avoiding higher doses. Expert opinion supports relative dose reductions for larger dogs, namely when using immunosuppressive doses of prednisolone, due to perceived increased risk of adverse effects [5]. Larger dogs have been shown to achieve higher blood concentrations than smaller dogs, when dosed by bodyweight, so its plausible that they would experience more pronounced side effects [24]. A recent retrospective study by Sri-Jayantha and colleagues (2022) supported this, finding increased bodyweight was associated with increased risk of muscle atrophy and polyphagia in dogs treated for IMHA and IMTP [23].

There are no previous studies evaluating risk factors for higher doses of prednisolone, or glucocorticoids in general, in dogs. Apart from bodyweight, other factors statistically significantly associated with prednisolone dose included age, sex/neuter status (for dosing in mg/kg, not mg/m$^2$), indicated dose and body system treated. Body system impacts the type of condition seen, with some body systems including predominantly inflammatory conditions and others immune mediated, particularly for the haematopoietic category which included the common immune mediated diseases and lymphoma. The finding that increases in age were associated with reduced prednisolone doses was unexpected. Dogs of younger age are more frequently diagnosed with immune mediated disease, perhaps leading to more frequent prescriptions of immunosuppressive doses [4]. However common inflammatory conditions, such as atopy, also tend to manifest when dogs are young [25]. Increasing age has been associated

with higher risk of polyuria and polydipsia as a side effect of glucocorticoids, potentially leading clinicians to be wary of higher doses in older dogs [8].

Data were acquired from VetCompass Australia participating practices and this may have resulted in a biased subset of the Australian veterinary clinic population. However, at the time of data acquisition, 137 practices contributed data to VetCompass Australia, thus the impact of this bias is likely to be small. As this study required manual data extraction from free text fields in the consultation records, it was impossible to include all eligible records. Thus a random sample of records was selected. The number of entries reviewed exceeded our sample size estimates so we believe that our analyses had sufficient power to identify characteristics associated with prednisolone dose, if they were in fact present. Our assessment of the clinician's indication for treatment was based entirely on the clarity of the medical record and this does allow for the potential for miscoding, if pertinent details that affected decision making were not included. We did not attempt to interpret the use of doses that were lower than the indicated dose we had assigned (such as a physiologic dose for an immune mediated disease), as tapering to the lowest effective dose is expected. Dose prescribed to each dog was also limited by what products are available and so tablet sizing may have influenced the exact dose the dog received. We did not describe how patient doses were tapered and so in cases where dogs received inappropriately high doses of prednisolone, we cannot comment on how long these doses were maintained.

## Conclusions

This study is the first assessing prednisolone prescribing practices in Australia in a large population of dogs. Use of VetCompass data provided a unique opportunity to describe prednisolone dose rates and to identify dog-level characteristics that influenced prednisolone dose rate. Prednisolone was primarily used as an anti-inflammatory, particularly for skin and ear disease. Inappropriate use of immunosuppressive doses of prednisolone for inflammatory conditions did occur but was the minority of prescriptions. Increasing bodyweight was associated with a small reduction in dose in mg/kg in this study, as well as when comparing Maltese dogs to Labrador dogs treated for inflammatory skin disease. This indicates that clinicians adjust prednisolone doses based on body size of their patient. Despite this, a large proportion of large dogs ($\geq$25 kg) receiving $\geq$50 mg/kg/day were dosed inappropriately. This emphasises the importance of ongoing communication of the newer guidelines for dosing of large breed dogs with prednisolone. More research around clinician awareness of dosing recommendations is needed, as well as a better understanding of prednisolone pharmacokinetics and pharmacodynamics to determine best practice for dosing for clinicians.

## Supporting information

**S1 Appendix. Body system categories for diagnoses, differentials or presenting complaints.**
(DOCX)

**S2 Appendix. Prednisolone indicated dose categories for diagnoses, differentials or presenting complaints.**
(DOCX)

**S3 Appendix. Prednisolone products prescribed to dogs in Australian veterinary practices.**
(DOCX)

**S1 File. Dog data (n = 2000 dogs).**
(XLSX)

**S2 File. Multivariable analysis for mg/kg and mg/m² dosing.**
(PDF)

**S3 File. Goodness of fit for multivariable analysis (mg/kg).**
(PDF)

**S4 File. Goodness of fit for multivariable analysis (mg/m²).**
(PDF)

## Acknowledgments

We would like to thank Mark Stevenson for his valuable input regarding our statistical analysis and for reviewing the manuscript.

## Author Contributions

**Conceptualization:** Bonnie Purcell, Anke Wiethoelter, Julien Dandrieux.

**Data curation:** Bonnie Purcell.

**Formal analysis:** Bonnie Purcell, Anke Wiethoelter.

**Funding acquisition:** Julien Dandrieux.

**Investigation:** Bonnie Purcell.

**Supervision:** Julien Dandrieux.

**Writing – original draft:** Bonnie Purcell.

**Writing – review & editing:** Anke Wiethoelter, Julien Dandrieux.

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
