## [Decision Letter · Decision Letter 0]

15 Dec 2022

PONE-D-22-28021Prednisolone prescribing practices for dogs in AustraliaPLOS ONE

Dear Dr. Purcell,

Thank you for submitting your manuscript to PLOS ONE. After careful consideration, we feel that it has merit but does not fully meet PLOS ONE’s publication criteria as it currently stands. Therefore, we invite you to submit a revised version of the manuscript that addresses the points raised during the review process. Specifically, please address concerns raised by Reviewer one about disease nomenclature for humans (Evans syndrome) versus other animals in veterinary medicine. Also please address the question concerning higher steroid doses in females, as well as clarify the two questions raised by reviewer two concerning discrepancies in dosing and the use of the term "physiologic" instead of "endocrine".

We look forward to receiving your revised manuscript.

Kind regards,

Mark Zabel

Academic Editor

PLOS ONE

Journal Requirements:

Reviewers' comments:

Reviewer's Responses to Questions

**Comments to the Author**

1. Is the manuscript technically sound, and do the data support the conclusions?

Reviewer #1: Yes

Reviewer #2: Yes

2. Has the statistical analysis been performed appropriately and rigorously? 

Reviewer #1: Yes

Reviewer #2: Yes

3. Have the authors made all data underlying the findings in their manuscript fully available?

Reviewer #1: Yes

Reviewer #2: Yes

4. Is the manuscript presented in an intelligible fashion and written in standard English?

Reviewer #1: Yes

Reviewer #2: Yes

5. Review Comments to the Author

Reviewer #1: This is a well written and appropriately powered study that appropriately addresses the topic of steroid prescribing practices in general practice. I have only two minor issues - Evans syndrome is mentioned in table 5 but is not defined in the text. Evans syndrome is generally considered to be a human term and in veterinary medicine it is preferred by most to not use the term Evans syndrome but rather to just label it as concurrent IMHA and IMTP. I would either avoid the term altogether or describe it in the text together with an explanation of why there is debate about use of this term in veterinary medicine. In the discussion you mention that intact female dogs were prescribed higher doses of steroids. Do you have any theories about why that might be?

Reviewer #2: Very well written and organized manuscript that made sound conclusions based on the data.

Two very small critics for authors consideration: 1. Anti inflammatory dose is defined in Table 3 as 0.3 <!--= 1.0 mg/kg/day but in line 219 the author refers to the publication dosage definition of 0.5-1 mg/kg/day. Although this difference is addressed in the conclusion the discrepancy provides confusion for the reader early on in the manuscript. The reviewer recommends line 219 reflect the Table 3 range rather than the publication (which was previously cited). <br /

2. On line 237 the author chose to use the word physiologic in the context of a disease indication when previous uses of physiologic were used to describe a dosing regime. It might be more clear for the author to use the disease indication defined in Table 3 ("endocrine").

6. PLOS authors have the option to publish the peer review history of their article (what does this mean?). If published, this will include your full peer review and any attached files.

Reviewer #1: **Yes: **Valerie Johnson DVM, PhD, DACVECC

Reviewer #2: No

---

## [Author Response · Author response to Decision Letter 0]

21 Jan 2023

To Mark Zabel and reviewers of “Prednisolone prescribing practices for dogs in Australia”, 

Thank you for considering our manuscript and the constructive feedback provided. Please find below our responses to the points raised during the review process. 

Reviewer #1, comment 1: We have changed “Evan’s syndrome” to “Concurrent immune mediated haemolytic anaemia and immune mediated thrombocytopenia” in Table 5. This is clearer and more correct. 

Reviewer #1, comment 2: There is no clear explanation for female entire dogs receiving higher doses than male neutered dogs in the mg/kg multivariable analysis. Noting that it is a relatively small increase, with a wide confidence interval, and that female entire dogs represent the smallest group (n = 114), we wanted to avoid over interpreting this result and so did not discuss it extensively. We have now added that this may be a type 1 error, given it was not found in the mg/m2 multivariable analysis. 

Reviewer #2, comment 1: We’re not quite sure, but think that this comment relates to the categories being not mutually exclusive in Table 3. Thank you for picking this up. We have revised the table and double-checked the text to ensure that the categories are now consistently defined as physiological (< 0.3 mg/kg), anti-inflammatory (0.3 – 1.0 mg/kg), intermediate (> 1.0 - < 1.5 mg/kg) and immunosuppressive (≥ 1.5 mg/kg) throughout the manuscript.

Reviewer #2, comment 2: Line 248 has been amended as suggested 

As an additional change, to improve clarity, we have changed the wording of one line in the discussion for clarity. This is line 385-386, regarding large dogs prescribed prednisolone at 50 mg/m2/day. The previous wording could have been misinterpreted and we think the new wording is clearer, pointing out that 67% of large dogs, receiving that dose, did so for inflammatory disease conditions. This can be reverted to the original wording if this is preferred. 

Finally, we have changed the address for the Melbourne Veterinary School to the Faculty of Science, due to a recent departmental change. 

Thank you again for your time and consideration, 

Bonnie Purcell, Julien Dandrieux and Anke Wiethoelter.

---

## [Decision Letter · Decision Letter 1]

15 Feb 2023

Prednisolone prescribing practices for dogs in Australia

PONE-D-22-28021R1

Dear Dr. Purcell,

We’re pleased to inform you that your manuscript has been judged scientifically suitable for publication and will be formally accepted for publication once it meets all outstanding technical requirements.

Kind regards,

Mark Zabel

Academic Editor

PLOS ONE

Additional Editor Comments (optional):

Reviewers' comments:

Reviewer's Responses to Questions

**Comments to the Author**

1. If the authors have adequately addressed your comments raised in a previous round of review and you feel that this manuscript is now acceptable for publication, you may indicate that here to bypass the “Comments to the Author” section, enter your conflict of interest statement in the “Confidential to Editor” section, and submit your "Accept" recommendation.

Reviewer #2: All comments have been addressed

2. Is the manuscript technically sound, and do the data support the conclusions?

Reviewer #2: Yes

3. Has the statistical analysis been performed appropriately and rigorously? 

Reviewer #2: Yes

4. Have the authors made all data underlying the findings in their manuscript fully available?

Reviewer #2: Yes

5. Is the manuscript presented in an intelligible fashion and written in standard English?

Reviewer #2: Yes

6. Review Comments to the Author

Reviewer #2: (No Response)

7. PLOS authors have the option to publish the peer review history of their article (what does this mean?). If published, this will include your full peer review and any attached files.

Reviewer #2: No

---

## [Editor Report · Acceptance letter]

20 Feb 2023

PONE-D-22-28021R1 

Prednisolone prescribing practices for dogs in Australia 

Dear Dr. Purcell:

I'm pleased to inform you that your manuscript has been deemed suitable for publication in PLOS ONE. Congratulations! Your manuscript is now with our production department. 

Kind regards, 

on behalf of

Dr. Mark Zabel 

Academic Editor

PLOS ONE